# Generalizing Across Domains via Cross-Gradient Training

Shiv Shankar[*], Vihari Piratla[*], Soumen Chakrabarti[1], Siddhartha Chaudhuri[1,2], Preethi Jyothi[1], and Sunita Sarawagi[1]

[1]Department of Computer Science, Indian Institute of Technology Bombay
[2]Adobe Research

## Abstract

We present CROSSGRAD, a method to use multi-domain training data to learn a classifier that generalizes to new domains. CROSSGRAD does not need an adaptation phase via labeled or unlabeled data, or domain features in the new domain. Most existing domain adaptation methods attempt to erase domain signals using techniques like domain adversarial training. In contrast, CROSSGRAD is free to use domain signals for predicting labels, if it can prevent overfitting on training domains. We conceptualize the task in a Bayesian setting, in which a sampling step is implemented as data augmentation, based on domain-guided perturbations of input instances. CROSSGRAD parallelly trains a label and a domain classifier on examples perturbed by loss gradients of each other's objectives. This enables us to directly perturb inputs, without separating and re-mixing domain signals while making various distributional assumptions. Empirical evaluation on three different applications where this setting is natural establishes that (1) domain-guided perturbation provides consistently better generalization to unseen domains, compared to generic instance perturbation methods, and that (2) data augmentation is a more stable and accurate method than domain adversarial training.

## 1 Introduction

We investigate how to train a classification model using multi-domain training data, so as to generalize to labeling instances from unseen domains. This problem arises in many applications, viz., handwriting recognition, speech recognition, sentiment analysis, and sensor data interpretation. In these applications, domains may be defined by fonts, speakers, writers, etc. Most existing work on handling a target domain not seen at training time requires either labeled or unlabeled data from the target domain at test time. Often, a separate "adaptation" step is then run over the source and target domain instances, only after which target domain instances are labeled. In contrast, we consider the situation where, during training, we have labeled instances from several domains which we can collectively exploit so that the trained system can handle new domains without the adaptation step.

### 1.1 Problem statement

Let $\mathcal{D}$ be a space of domains. During training we get labeled data from a proper subset $D \subset \mathcal{D}$ of these domains. Each labeled example during training is a triple $(\mathbf{x}, y, d)$ where $\mathbf{x}$ is the input, $y \in \mathcal{Y}$ is the true class label from a finite set of labels $\mathcal{Y}$ and $d \in D$ is the domain from which this example is sampled. We must train a classifier to predict the label $y$ for examples sampled from all domains, including the subset $\mathcal{D} \setminus D$ not seen in the training set. Our goal is high accuracy for both in-domain (i.e., in $D$) and out-of-domain (i.e., in $\mathcal{D} \setminus D$) test instances.

One challenge in learning a classifier that generalizes to unseen domains is that $\Pr(y|\mathbf{x})$ is typically harder to learn than $\Pr(y|\mathbf{x}, d)$. While Yang & Hospedales (2015) addressed a similar setting, they assumed a specific geometry characterizing the domain, and performed kernel regression in

---

[*]These two authors contributed equally

this space. In contrast, in our setting, we wish to avoid any such explicit domain representation, appealing instead to the power of deep networks to discover implicit features.

Lacking any feature-space characterization of the domain, conventional training objectives (given a choice of hypotheses having sufficient capacity) will tend to settle to solutions that overfit on the set of domains seen during training. A popular technique in the domain adaptation literature to generalize to new domains is domain adversarial training (Ganin et al., 2016; Tzeng et al., 2017). As the name suggests, here the goal is to learn a transformation of input $\mathbf{x}$ to a domain-independent representation, with the hope that amputating domain signals will make the system robust to new domains. We show in this paper that such training does not necessarily safeguard against over-fitting of the network as a whole. We also argue that even if such such overfitting could be avoided, we do not necessarily want to wipe out domain signals, if it helps in-domain test instances.

## 1.2 CONTRIBUTION

In a marked departure from domain adaptation via amputation of domain signals, we approach the problem using a form of data augmentation based on domain-guided perturbations of input instances. If we could model exactly how domain signals for $d$ manifest in $\mathbf{x}$, we could simply replace these signals with those from suitably sampled other domains $d'$ to perform data augmentation. We first conceptualize this in a Bayesian setting: discrete domain $d$ 'causes' continuous multivariate $\mathbf{g}$, which, in combination with $y$, 'causes' $\mathbf{x}$. Given an instance $\mathbf{x}$, if we can recover $\mathbf{g}$, we can then perturb $\mathbf{g}$ to $\mathbf{g}'$, thus generating an augmented instance $\mathbf{x}'$. Because such perfect domain perturbation is not possible in reality, we first design an (imperfect) domain classifier network to be trained with a suitable loss function. Given an instance $\mathbf{x}$, we use the loss gradient w.r.t. $\mathbf{x}$ to perturb $\mathbf{x}$ in directions that change the domain classifier loss the most. The training loss for the $y$-predictor network on original instances is combined with the training loss on the augmented instances. We call this approach *cross-gradient training*, which is embodied in a system we describe here, called CROSSGRAD. We carefully study the performance of CROSSGRAD on a variety of domain adaptive tasks: character recognition, handwriting recognition and spoken word recognition. We demonstrate performance gains on new domains without any out-of-domain instances available at training time.

## 2 RELATED WORK

Domain adaptation has been studied under many different settings: two domains (Ganin et al., 2016; Tzeng et al., 2017) or multiple domains (Mansour et al., 2009; Zhang et al., 2015), with target domain data that is labeled (III, 2007; III et al., 2010; Saenko et al., 2010) or unlabeled (Gopalan et al., 2011; Gong et al., 2012; Ganin et al., 2016), paired examples from source and target domain (Kuan-Chuan et al., 2017), or domain features attached with each domain (Yang & Hospedales, 2016). Domain adaptation techniques have been applied to numerous tasks in speech, language processing and computer vision (Woodland, 2001; Saon et al., 2013; Jiang & Zhai, 2007; III, 2007; Saenko et al., 2010; Gopalan et al., 2011; Li et al., 2013; Huang & Belongie, 2017; Upchurch et al., 2016). However, unlike in our setting, these approaches typically assume the availability of some target domain data which is either labeled or unlabeled.

For neural networks a recent popular technique is domain adversarial networks (DANs) (Tzeng et al., 2015; 2017; Ganin et al., 2016). The main idea of DANs is to learn a representation in the last hidden layer (of a multilayer network) that cannot discriminate among different domains in the input to the first layer. A domain classifier is created with the last layer as input. If the last layer encapsulates no domain information apart from what can be inferred from the label, the accuracy of the domain classifier is low. The DAN approach makes sense when all domains are visible during training. In this paper, our goal is to generalize to unseen domains.

Domain generalization is traditionally addressed by learning representations that encompass information from all the training domains. Muandet et al. (2013) learn a kernel-based representation that minimizes domain dissimilarity and retains the functional relationship with the label. Gan et al. (2016) extends Muandet et al. (2013) by exploiting attribute annotations of examples to learn new feature representations for the task of attribute detection. In Ghifary et al. (2015), features that are shared across several domains are estimated by jointly learning multiple data-reconstruction tasks.

Such representations are shown to be effective for domain generalization, but ignore any additional information that domain features can provide about labels.

Domain adversarial networks (DANs) (Ganin et al., 2016) can also be used for domain generalization in order to learn domain independent representations. A limitation of DANs is that they can be misled by a representation layer that over-fits to the set of training domains. In the extreme case, a representation that simply outputs label logits via a last linear layer (making the softmax layer irrelevant) can keep both the adversarial loss and label loss small, and yet not be able to generalize to new test domains. In other words, not being able to infer the domain from the last layer does not imply that the classification is domain-robust.

Since we do not assume any extra information about the test domains, conventional approaches for regularization and generalizability are also relevant. Xu et al. (2014) use exemplar-based SVM classifiers regularized by a low-rank constraint over predictions. Khosla et al. (2012) also deploy SVM based classifier and regularize the domain specific components of the learners. The method most related to us is the adversarial training of (Szegedy et al., 2013; Goodfellow et al., 2014; Miyato et al., 2016) where examples perturbed along the gradient of classifier loss are used to augment the training data. perturbs examples. Instead, our method attempts to model domain variation in a continuous space and perturbs examples along domain loss.

Our Bayesian model to capture the dependence among label, domains, and input is similar to Zhang et al. (2015, Fig. 1d), but the crucial difference is the way the dependence is modeled and estimated. Our method attempts to model domain variation in a continuous space and project perturbation in that space to the instances.

## 3 OUR APPROACH

We assume that input objects are characterized by two uncorrelated or weakly correlated tags: their *label* and their *domain*. E.g. for a set of typeset characters, the label could be the corresponding character of the alphabet ('A', 'B' etc) and the domain could be the font used to draw the character. In general, it should be possible to change any one of these, while holding the other fixed.

We use a Bayesian network to model the dependence among the label $y$, domain $d$, and input $\mathbf{x}$ as shown in Figure 1. Variables $y \in \mathcal{Y}$ and $d \in \mathcal{D}$ are discrete and $\mathbf{g} \in \mathbb{R}^q, \mathbf{x} \in \mathbb{R}^r$ lie in continuous multi-dimensional spaces. The domain $d$ induces a set of latent domain features $\mathbf{g}$. The input $\mathbf{x}$ is

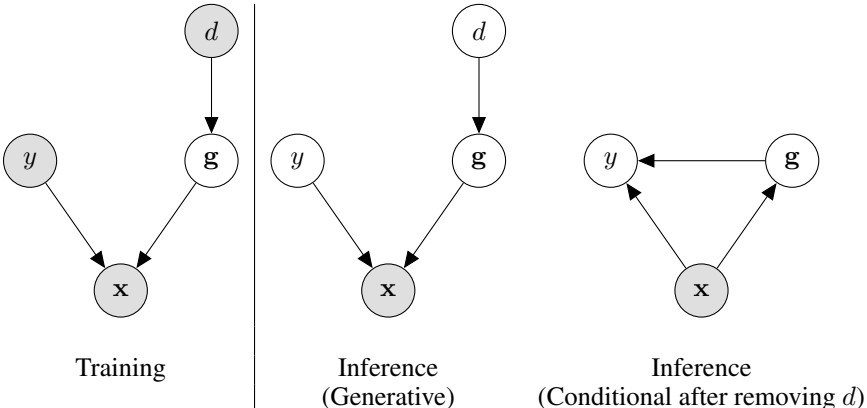

|  Training | Inference (Generative) | Inference (Conditional after removing $d$) |

Figure 1: Bayesian network to model dependence between label $y$, domain $d$, and input $\mathbf{x}$.

obtained by a complicated, un-observed mixing[1] of $y$ and $\mathbf{g}$. In the training sample $L$, nodes $y, d, \mathbf{x}$ are observed but $L$ spans only a proper subset $D$ of the set of all domains $\mathcal{D}$. During inference, only $\mathbf{x}$ is observed and we need to compute the posterior $\Pr(y|\mathbf{x})$. As reflected in the network, $y$ is not independent of $d$ given $\mathbf{x}$. However, since $d$ is discrete and we observe only a subset of $d$'s during

---

[1]The dependence of $y$ on $\mathbf{x}$ could also be via continuous hidden variables but our model for domain generalization is agnostic of such structure.

training, we need to make additional assumptions to ensure that we can generalize to a new $d$ during testing. The assumption we make is that integrated over the training domains the distribution $P(\mathbf{g})$ of the domain features is well-supported in $L$. More precisely, generalizability of a training set of domains $D$ to the universe of domains $\mathcal{D}$ requires that

$$P(\mathbf{g}) = \sum_{d \in \mathcal{D}} P(\mathbf{g}|d)P(d) \approx \sum_{d \in D} P(\mathbf{g}|d)\frac{P(d)}{P(D)} \tag{A1}$$

Under this assumption $P(\mathbf{g})$ can be modeled during training, so that during inference we can infer $y$ for a given $\mathbf{x}$ by estimating

$$\Pr(y|\mathbf{x}) = \sum_{d \in \mathcal{D}} \Pr(y|\mathbf{x}, d)\Pr(d|\mathbf{x}) = \int_{\mathbf{g}} \Pr(y|\mathbf{x}, \mathbf{g})\Pr(\mathbf{g}|\mathbf{x}) \approx \Pr(y|\mathbf{x}, \hat{\mathbf{g}}) \tag{1}$$

where $\hat{\mathbf{g}} = \text{argmax}_{\mathbf{g}} \Pr(\mathbf{g}|\mathbf{x})$ is the inferred continuous representation of the domain of $\mathbf{x}$.

This assumption is key to our being able to claim generalization to new domains even though most real-life domains are discrete. For example, domains like fonts and speakers are discrete, but their variation can be captured via latent continuous features (e.g. slant, ligature size etc. of fonts; speaking rate, pitch, intensity, etc. for speech). The assumption states that as long as the training domains span the latent continuous features we can generalize to new fonts and speakers.

We next elaborate on how we estimate $\Pr(y|\mathbf{x}, \hat{\mathbf{g}})$ and $\hat{\mathbf{g}}$ using the domain labeled data $L = \{(\mathbf{x}, y, d)\}$. The main challenge in this task is to ensure that the model for $\Pr(y|\mathbf{x}, \hat{\mathbf{g}})$ is not overfitted on the inferred $\mathbf{g}$'s of the training domains. In many applications, the per-domain $\Pr(y|\mathbf{x}, d)$ is significantly easier to train. So, an easy local minima is to choose a different $\mathbf{g}$ for each training $d$ and generate separate classifiers for each distinct training domain. We must encourage the network to stay away from such easy solutions. We strive for generalization by moving along the continuous space $\mathbf{g}$ of domains to sample new training examples from hallucinated domains. Ideally, for each training instance $(\mathbf{x}_i, y_i)$ from a given domain $d_i$, we wish to generate a new $\mathbf{x}'$ by transforming its (inferred) domain $\mathbf{g}_i$ to a random domain sampled from $P(\mathbf{g})$, keeping its label $y_i$ unchanged. Under the domain continuity assumption (A1), a model trained with such an ideally augmented dataset is expected to generalize to domains in $\mathcal{D} \setminus D$.

However, there are many challenges to achieving such ideal augmentation. To avoid changing $y_i$, it is convenient to draw a sample $\mathbf{g}$ by perturbing $\mathbf{g}_i$. But $\mathbf{g}_i$ may not be reliably inferred, leading to a distorted sample of $\mathbf{g}$. For example, if the $\mathbf{g}_i$ obtained from an imperfect extraction conceals label information, then big jumps in the approximate $\mathbf{g}$ space could change the label too. We propose a more cautious data augmentation strategy that perturbs the input to make only small moves along the estimated domain features, while changing the label as little as possible. We arrive at our method as follows.

**Domain inference.** We create a model $G(\mathbf{x})$ to extract domain features $\mathbf{g}$ from an input $\mathbf{x}$. We supervise the training of $G$ to predict the domain label $d_i$ as $S(G(\mathbf{x}_i))$ where $S$ is a softmax transformation. We use $J_d$ to denote the cross-entropy loss function of this classifier. Specifically, $J_d(\mathbf{x}_i, d_i)$ is the domain loss at the current instance.

**Domain perturbation.** Given an example $(\mathbf{x}_i, y_i, d_i)$, we seek to sample a new example $(\mathbf{x}'_i, y_i)$ (i.e., with the same label $y_i$), whose domain is as "far" from $d_i$ as possible. To this end, consider setting $\mathbf{x}'_i = \mathbf{x}_i + \epsilon \nabla_{\mathbf{x}_i} J_d(\mathbf{x}_i, d_i)$. Intuitively, this perturbs the input along the direction of greatest domain change[2], for a given budget of $||\mathbf{x}'_i - \mathbf{x}_i||$. However, this approach presupposes that the direction of domain change in our domain classifier is not highly correlated with the direction of label change. To enforce this

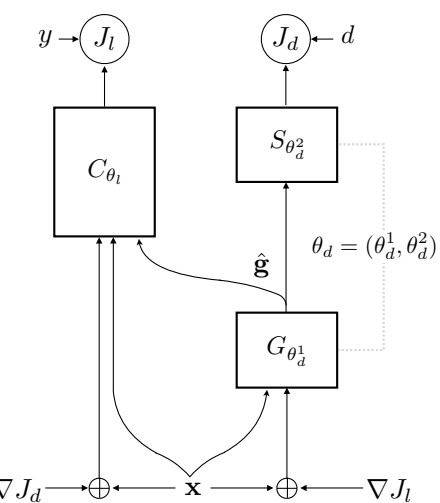

Figure 2: CROSSGRAD network design.

---

[2]We use $\nabla_{\mathbf{x}_i} J_d$ as shorthand for the gradient $\nabla_{\mathbf{x}} J_d$ evaluated at $\mathbf{x}_i$.

in our model, we shall train the domain feature extractor $G$ to avoid domain shifts when the data is perturbed to cause label shifts.

What is the consequent change of the continuous domain features $\hat{\mathbf{g}}_i$? This turns out to be $\epsilon \mathbb{J}\mathbb{J}^\top \nabla_{\hat{\mathbf{g}}_i} J_d(\mathbf{x}_i, d_i)$, where $\mathbb{J}$ is the Jacobian of $\hat{\mathbf{g}}$ w.r.t. $\mathbf{x}$. Geometrically, the $\mathbb{J}\mathbb{J}^\top$ term is the (transpose of the) metric tensor matrix accounting for the distortion in mapping from the $\mathbf{x}$-manifold to the $\hat{\mathbf{g}}$-manifold. While this perturbation is not very intuitive in terms of the direct relation between $\hat{\mathbf{g}}$ and $d$, we show in the Appendix that the input perturbation $\epsilon \nabla_{\mathbf{x}_i} J_d(\mathbf{x}_i, d_i)$ is *also* the first step of a gradient descent process to induce the "natural" domain perturbation $\Delta\hat{\mathbf{g}}_i = \epsilon' \nabla_{\hat{\mathbf{g}}_i} J_d(\mathbf{x}_i, d_i)$.

The above development leads to the network sketched in Figure 2, and an accompanying training algorithm, CROSSGRAD, shown in Algorithm 1. Here $X, Y, D$ correspond to a minibatch of instances. Our proposed method integrates data augmentation and batch training as an alternating sequence of steps. The domain classifier is simultaneously trained with the perturbations from the label classifier network so as to be robust to label changes. Thus, we construct cross-objectives $J_l$ and $J_d$, and update their respective parameter spaces. We found this scheme of simultaneously training both networks to be empirically superior to independent training even though the two classifiers do not share parameters.

---

**Algorithm 1** CROSSGRAD training pseudocode.

---

1: **Input:** Labeled data $\{(\mathbf{x}_i, y_i, d_i)\}_{i=1}^M$, step sizes $\epsilon_l, \epsilon_d$, learning rate $\eta$, data augmentation weights $\alpha_l, \alpha_d$, number of training steps $n$.
2: **Output:** Label and domain classifier parameters $\theta_l, \theta_d$
3: Initialize $\theta_l, \theta_d$ $\{J_l, J_d$ are loss functions for the label and domain classifiers, respectively$\}$
4: **for** $n$ training steps **do**
5:     Sample a labeled batch $(X, Y, D)$
6:     $X_d := X + \epsilon_l \cdot \nabla_X J_d(X, D; \theta_d)$
7:     $X_l := X + \epsilon_d \cdot \nabla_X J_l(X, Y; \theta_l)$
8:     $\theta_l \leftarrow \theta_l - \eta \nabla_{\theta_l}((1 - \alpha_l) J_l(X, Y; \theta_l) + \alpha_l J_l(X_d, Y; \theta_l))$
9:     $\theta_d \leftarrow \theta_d - \eta \nabla_{\theta_d}((1 - \alpha_d) J_d(X, D; \theta_d) + \alpha_d J_d(X_l, D; \theta_d))$
10: **end for**

---

If $y$ and $d$ are completely correlated, CROSSGRAD reduces to traditional adversarial training. If, on the other extreme, they are perfectly uncorrelated, removing domain signal should work well. The interesting and realistic situation is where they are only partially correlated. CROSSGRAD is designed to handle the whole spectrum of correlations.

## 4 EXPERIMENTS

In this section, we demonstrate that CROSSGRAD provides effective domain generalization on four different classification tasks under three different model architectures. We provide evidence that our Bayesian characterization of domains as continuous features is responsible for such generalization. We establish that CROSSGRAD's domain guided perturbations provide a more consistent generalization to new domains than label adversarial perturbation (Goodfellow et al., 2014) which we denote by LABELGRAD. Also, we show that DANs, a popular domain adaptation method that suppresses domain signals, provides little improvement over the baseline (Ganin et al., 2016; Tzeng et al., 2017).

We describe the four different datasets and present a summary in Table 3.

**Character recognition across fonts.** We created this dataset from Google Fonts[3]. The task is to identify the character across different fonts as the domain. The label set consists of twenty-six letters of the alphabet and ten numbers. The data comprises of 109 fonts which are partitioned as 65% train, 25% test and 10% validation folds. For each font and label, two to eighteen images are generated by randomly rotating the image around the center and adding pixel-level random noise. The neural network is LeNet (LeCun et al., 1998) modified to have three pooled convolution layers instead of two and ReLU activations instead of tanh.

---

[3] https://fonts.google.com/?category=Handwriting&subset=latin

| Dataset | Label | Domain |
|---------|-------|--------|
| Font | 36 characters | 109 fonts |
| Handwriting | 111 characters | 74 authors |
| MNIST | 10 digits | 6 rotations |
| Speech | 12 commands | 1888 speakers |

Table 3: Summary of datasets.

**Handwriting recognition across authors.**   We used the LipiTk dataset that comprises of hand-written characters from the Devanagari script[4]. Each writer is considered as a domain, and the task is to recognize the character. The images are split on writers as 60% train, 25% test and 15% validation. The neural network is the same as for the Fonts dataset above.

**MNIST across synthetic domains.**   This dataset derived from MNIST was introduced by Ghifary et al. (2015). Here, labels comprise the 10 digits and domains are created by rotating the images in multiples of 15 degrees: 0, 15, 30, 45, 60 and 75. The domains are labeled with the angle by which they are rotated, e.g., M15, M30, M45. We tested on domain M15 while training on the rest. The network is the 2-layer convolutional one used by Motiian et al. (2017).

**Spoken word recognition across users.**   We used the Google Speech Command Dataset[5] that consists of spoken word commands from several speakers in different acoustic settings. Spoken words are labels and speakers are domains. We used 20% of domains each for testing and validation. The number of training domains was 100 for the experiments in Table 4. We also report performance with varying numbers of domains later in Table 8. We use the architecture of Sainath & Parada (2015)[6].

For all experiments, the set of domains in the training, test, and validation sets were disjoint. We selected hyper-parameters based on accuracy on the validation set as follows. For LABELGRAD the parameter $\alpha$ was chosen from $\{0.1, 0.25, 0.75, 0.5, 0.9\}$ and for CROSSGRAD we chose $\alpha_l = \alpha_d$ from the same set of values. We chose $\epsilon$ ranges so that $L_\infty$ norm of the perturbations are of similar sizes in LABELGRAD and CROSSGRAD. The multiples in the $\epsilon$ range came from $\{0.5, 1, 2, 2.5\}$. The optimizer for the first three datasets is RMS prop with a learning rate ($\eta$) of 0.02 whereas for the last Speech dataset it is SGD with $\eta = 0.001$ initially and 0.0001 after 15 iterations. In CROSSGRAD networks, **g** is incorporated in the label classifier network by concatenating with the output from the last but two hidden layer.

## 4.1 OVERALL COMPARISON

In Table 4 we compare CROSSGRAD with domain adversarial networks (DAN), label adversarial perturbation (LABELGRAD), and a baseline that performs no special training. For the MNIST dataset the baseline is CCSA (Motiian et al., 2017) and D-MTAE (Ghifary et al., 2015). We observe that, for all four datasets, CROSSGRAD provides an accuracy improvement. DAN, which is designed specifically for domain adaptation, is worse than LABELGRAD, which does not exploit domain signal in any way. While the gap between LABELGRAD and CROSSGRAD is not dramatic, it is consistent as supported by this table and other experiments that we later describe.

| Method Name | Fonts | Handwriting | MNIST | Speech |
|-------------|-------|-------------|-------|--------|
| Baseline | 68.5 | 82.5 | 95.6 | 72.6 |
| DAN | 68.9 | 83.8 | 98.0 | 70.4 |
| LABELGRAD | 71.4 | 86.3 | 97.8 | 72.7 |
| CROSSGRAD | **72.6** | **88.6** | **98.6** | **73.5** |

Table 4:  Accuracy on four different datasets. The baseline for MNIST is CCSA.

**Changing model architecture.**   In order to make sure that these observed trends hold across model architectures, we compare different methods with the model changed to a 2-block ResNet (He et al.,

---

[4] http://lipitk.sourceforge.net/hpl-datasets.htm

[5] https://research.googleblog.com/2017/08/launching-speech-commands-dataset.html

[6] https://www.tensorflow.org/versions/master/tutorials/audio_recognition

2016) instead of LeNet (LeCun et al., 1998) for the Fonts and Handwriting dataset in Table 5. For both datasets the ResNet model is significantly better than the LeNet model. But even for the higher capacity ResNet model, CROSSGRAD surpasses the baseline accuracy as well as other methods like LABELGRAD .

| Method Name | Fonts | | Handwriting | |
| --- | --- | --- | --- | --- |
| | LeNet | ResNet | LeNet | ResNet |
| Baseline | 68.5 | 80.2 | 82.5 | 91.5 |
| DAN | 68.9 | 81.1 | 83.8 | 88.5 |
| LABELGRAD | 71.4 | 80.5 | 86.3 | 91.8 |
| CROSSGRAD | **72.6** | **82.4** | **88.6** | **92.1** |

Table 5: Accuracy with varying model architectures.

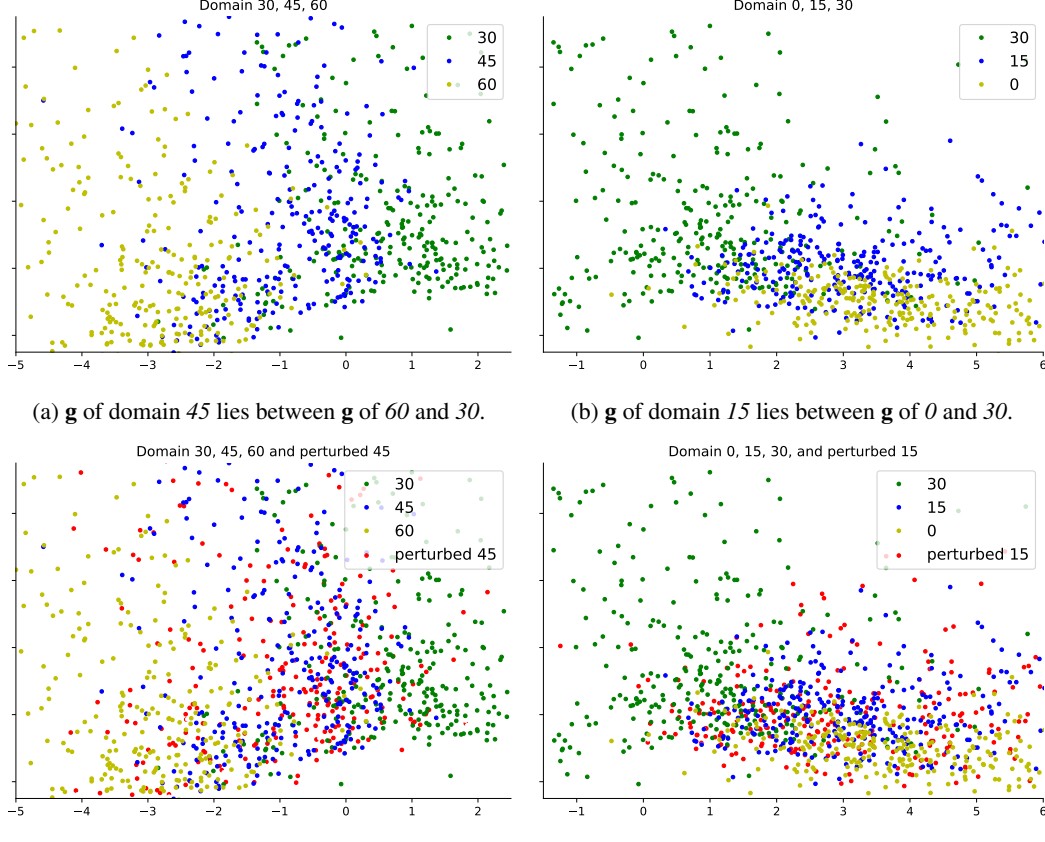

(a) **g** of domain *45* lies between **g** of *60* and *30*.     (b) **g** of domain *15* lies between **g** of *0* and *30*.

(c) Adding **g** of perturbed domain *45* to above.     (d) Adding **g** of perturbed domain *15* to above.

Figure 6: Comparing domain embeddings (**g**) across domains. Each color denotes a domain.

## 4.2 WHY DOES CROSSGRAD WORK?

We present insights on the working of CROSSGRAD via experiments on the MNIST dataset where the domains corresponding to image rotations are easy to interpret.

In Figure 6a we show PCA projections of the **g** embeddings for images from three different domains, corresponding to rotations by *30, 45, 60* degrees in green, blue, and yellow respectively. The **g** embeddings of domain *45* (blue) lies in between the **g** of domains *30* (green) and *60* (yellow) showing that the domain classifier has successfully extracted continuous representation of the domain even when the input domain labels are categorical. Figure 6b shows the same pattern for domains *0,*

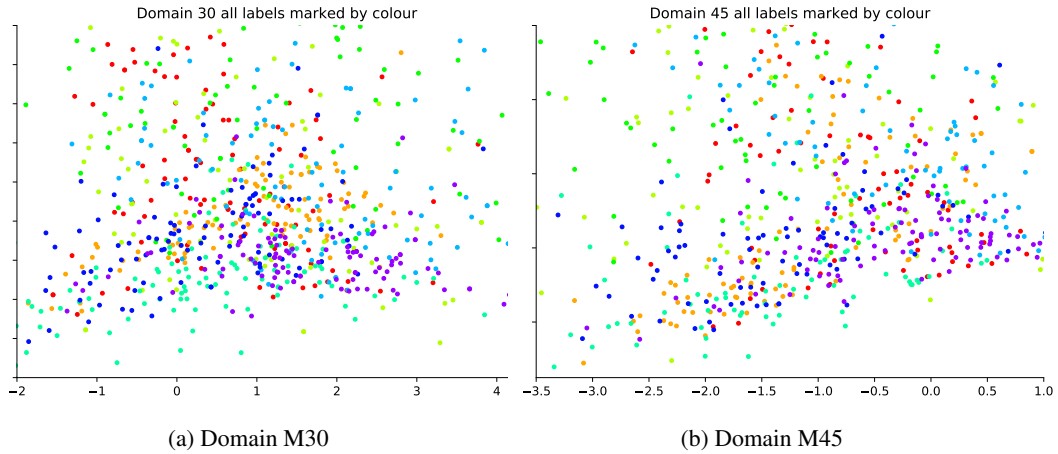

(a) Domain M30             (b) Domain M45

Figure 7: Comparing domain embeddings (**g**) across labels. Each color denotes a label. Unclustered colors indicates that label information is mostly absent from **g**.

*15, 30.* Here again we see that the embedding of domain *15* (blue) lies in-between that of domain *0* (yellow) and *30* (green).

Next, we show that the **g** perturbed along gradients of domain loss, does manage to generate images that substitute for the missing domains during training. For example, the embeddings of the domain *45*, when perturbed, scatters towards the domain *30* and *60* as can be seen in Figure 6c: note the scatter of *perturbed 45* (red) points inside the *30* (green) zone, without any *45* (blue) points. Figure 6d depicts a similar pattern with perturbed domain embedding points (red) scattering towards domains *30* and *0* more than unperturbed domain *15* (blue). For example, between x-axis -1 and 1 dominated by the green domain (domain *30*) we see many more red points (perturbed domain *15*) than blue points (domain *15*). Similarly in the lower right corner of domain *0* shown in yellow. This highlights the mechanism of CROSSGRAD working; that it is able to augment training with samples closer to unobserved domains.

Finally, we observe in Figure 7 that the embeddings are not correlated with labels. For both domains *30* and *45* the colors corresponding to different labels are not clustered. This is a consequence of CROSSGRAD's symmetric training of the domain classifier via label-loss perturbed images.

### 4.3 WHEN IS DOMAIN GENERALIZATION EFFECTIVE?

We next present a couple of experiments that provide insight into the settings in which CROSSGRAD is most effective.

First, we show the effect of increasing the number of training domains. Intuitively, we expect CROSSGRAD to be most useful when training domains are scarce and do not directly cover the test domains. We verified this on the speech dataset where the number of available domains is large. We varied the number of training domains while keeping the test and validation data fixed. Table 8 summarizes our results. Note that CROSSGRAD outperforms the baseline and LABELGRAD most significantly when the number of training domains is small (40). As the training data starts to cover more and more of the possible domain variations, the marginal improvement provided by CROSS-GRAD decreases. In fact, when the models are trained on the full training data (consisting of more than 1000 domains), the baseline achieves an accuracy of 88.3%, and both CROSSGRAD and LA-BELGRAD provide no gains[7] beyond that. DAN, like in other datasets, provides unstable gains and is difficult to tune. LABELGRAD shows smaller relative gains than CROSSGRAD but follows the same trend of reducing gains with increasing number of domains.

In general, how CROSSGRAD handles multidimensional, non-linear involvement of **g** in determining **x** is difficult to diagnose. To initiate a basic understanding of how data augmentation supplies CROSSGRAD with hallucinated domains, we considered a restricted situation where the discrete domain is secretly a continuous 1-d space, namely, the angle of rotation in MNIST. In this setting,

---

[7]The gap in accuracy between the baseline and CROSSGRAD for the case of 1000 domains is not statistically significant according to the MAPSSWE test (Gillick & Cox, 1989).

| Method Name | 40 domains | 100 domains | 200 domains | 1000 domains |
|---|---|---|---|---|
| Baseline | 62.2 | 72.6 | 79.1 | 88.3 |
| LABELGRAD | +1.4 | +0.1 | -0.1 | -0.7 |
| DAN | +0.4 | -2.2 | +0.7 | -3.3 |
| CROSSGRAD | +2.3 | +0.9 | +0.7 | -0.4 |

Table 8: Accuracy on Google Speech Command Task. For each method we show the relative improvement (positive or negative) over the baseline.

a natural question is, given a set of training domains (angles), which test domains (angles) perform well under CROSSGRAD?

| Test domains → | M0 | M15 | M30 | M45 | M60 | M75 |
|---|---|---|---|---|---|---|
| CCSA | 84.6 | 95.6 | 94.6 | 82.9 | 94.8 | 82.1 |
| D-MTAE | 82.5 | 96.3 | 93.4 | 78.6 | 94.2 | 80.5 |
| LABELGRAD | **89.7** | 97.8 | 98.0 | 97.1 | 96.6 | **92.1** |
| DAN | 86.7 | 98.0 | 97.8 | 97.4 | 96.9 | 89.1 |
| CROSSGRAD | 88.3 | **98.6** | **98.0** | **97.7** | **97.7** | 91.4 |

Table 9: Accuracy on rotated MNIST.

We conducted leave-one-domain-out experiments by picking one domain as the test domain, and providing the others as training domains. In Table 9 we compare the accuracy of different methods. We also compare against the numbers reported by the CCSA method of domain generalization (Motiian et al., 2017) as reported by the authors.

It becomes immediately obvious from Table 9 that CROSSGRAD is beaten in only two cases: M0 and M75, which are the two extreme rotation angles. For angles in the middle, CROSSGRAD is able to interpolate the necessary domain representation **g** via 'hallucination' from other training domains. Recall from Figures 6c and 6d that the perturbed **g** during training covers for the missing test domains. In contrast, when M0 or M75 are in the test set, CROSSGRAD's domain loss gradient does not point in the direction of domains outside the training domains. If and how this insight might generalize to more dimensions or truly categorical domains is left as an open question.

## 5  CONCLUSION

Domain $d$ and label $y$ interact in complicated ways to influence the observable input **x**. Most domain adaption strategies implicitly consider the domain signal to be extraneous and seek to remove its effect to train more robust label predictors. We presented CROSSGRAD, which considers them in a more symmetric manner. CROSSGRAD provides a new data augmentation scheme based on the $y$ (respectively, $d$) predictor using the gradient of the $d$ (respectively, $y$) predictor over the input space, to generate perturbations. Experiments comparing CROSSGRAD with various recent adversarial paradigms show that CROSSGRAD can make better use of partially correlated $y$ and $d$, without requiring explicit distributional assumptions about how they affect **x**. CROSSGRAD is at its best when training domains are scarce and do not directly cover test domains well. Future work includes extending CROSSGRAD to exploit labeled or unlabeled data in the test domain, and integrating the best of LABELGRAD and CROSSGRAD into a single algorithm.

ACKNOWLEDGEMENTS

We gratefully acknowledge the support of NVIDIA Corporation with the donation of Titan X GPUs used for this research. We thank Google for supporting travel to the conference venue.

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

## APPENDIX

**Relating the "natural" perturbations of x and $\hat{\mathbf{g}}$.** In Section 3, we claimed that the intuitive perturbation $\epsilon \nabla_{\mathbf{x}_i} J_d(\mathbf{x}_i, d_i)$ of $\mathbf{x}_i$ attempts to induce the intuitive perturbation $\epsilon' \nabla_{\hat{\mathbf{g}}_i} J_d(\mathbf{x}_i, d_i)$ of $\hat{\mathbf{g}}_i$, even though the exact relation between a perturbation of $\mathbf{x}_i$ and that of $\hat{\mathbf{g}}_i$ requires the metric tensor transpose $\mathbb{J}\mathbb{J}^\top$. We will now prove this assertion. Of course, an isometric map $G : \mathbf{x} \mapsto \mathbf{g}$, with an orthogonal Jacobian ($\mathbb{J}^{-1} = \mathbb{J}^\top$), trivially has this property, but we present an alternative derivation which may give further insight into the interaction between the perturbations in the general case.

Consider perturbing $\hat{\mathbf{g}}_i$ to produce $\mathbf{g}'_i = \hat{\mathbf{g}}_i + \epsilon' \nabla_{\hat{\mathbf{g}}_i} J_d(\mathbf{x}_i, d_i)$. This yields a new augmented input instance $\mathbf{x}'_i$ as

$$G^{-1}(\mathbf{g}'_i) = G^{-1}\big(G(\mathbf{x}_i) + \epsilon' \nabla_{\hat{\mathbf{g}}_i} J_d(\mathbf{x}_i, d_i)\big).$$

We show next that the perturbed $\mathbf{x}'_i$ can be approximated by $\mathbf{x}_i + \epsilon \nabla_{\mathbf{x}_i} J_d(\mathbf{x}_i, d_i)$.

In this proof we drop the subscript $i$ for ease of notation. In the forward direction, the relationship between $\Delta \mathbf{x}$ and $\Delta \hat{\mathbf{g}}$ can be expressed using the Jacobian $\mathbb{J}$ of $\hat{\mathbf{g}}$ w.r.t. $\mathbf{x}$:

$$\Delta \hat{\mathbf{g}} = \mathbb{J} \Delta \mathbf{x}$$

To invert the relationship for a non-square and possibly low-rank Jacobian, we use the Jacobian transpose method devised for inverse kinematics (Balestrino et al., 1984; Wolovich & Elliott, 1984). Specifically, we write $\Delta \hat{\mathbf{g}} = \hat{\mathbf{g}}(\mathbf{x}') - \hat{\mathbf{g}}(\mathbf{x})$, and recast the problem as trying to minimize the squared L2 error

$$L_{\hat{\mathbf{g}}}(\mathbf{x}) = \tfrac{1}{2} \left(\hat{\mathbf{g}}(\mathbf{x}') - \hat{\mathbf{g}}(\mathbf{x})\right)^\top \left(\hat{\mathbf{g}}(\mathbf{x}') - \hat{\mathbf{g}}(\mathbf{x})\right)$$

with gradient descent. The gradient of the above expression w.r.t. $\mathbf{x}$ is

$$\nabla_{\mathbf{x}} L_{\hat{\mathbf{g}}} \quad = \quad -\left(\Delta \hat{\mathbf{g}}^\top \frac{\partial \hat{\mathbf{g}}}{\partial \mathbf{x}}\right)^\top \quad = \quad -\mathbb{J}^\top \Delta \hat{\mathbf{g}}$$

Hence, the initial gradient descent step to affect a change of $\Delta \hat{\mathbf{g}}$ in the domain features would increment $\mathbf{x}$ by $\epsilon \mathbb{J}^{\top} \Delta \hat{\mathbf{g}}$. The Jacobian, which is a matrix of first partial derivatives, can be computed by back-propagation. Thus we get

$$\Delta \mathbf{x} = \epsilon \mathbb{J}^{\top} \nabla_{\hat{\mathbf{g}}} J_d(\hat{\mathbf{g}}, d) = \epsilon \sum_i \frac{\partial \hat{g}^i}{\partial \mathbf{x}}^{\top} \frac{\partial J_d(\hat{\mathbf{g}}, d)}{\partial \hat{g}^i},$$

which, by the chain rule, gives

$$\Delta \mathbf{x} = \epsilon \nabla_{\mathbf{x}} J_d(\mathbf{x}, d).$$

