# OpenReview forum: "Generalizing Across Domains via Cross-Gradient Training"
_ICLR.cc/2018/Conference — Accept (Poster)_

### Official Review · AnonReviewer1 · 2017-11-27
**Novel approach and extensive experiments. Many related works are missing.**

**Rating:** 7
**Confidence:** 5

**Review:**

This paper proposed a domain generalization approach by domain-dependent data augmentation. The augmentation is guided by a network that is trained to classify a data point to different domains. Experiments on four datasets verify the effectiveness of the proposed approach.

Strengths:
+ The proposed classification model is domain-dependent, as opposed to being domain-invariant. This is new and differs from most existing works on domain adaptation/generalization, to the best of my knowledge.
+ The experiments show that the proposed method outperforms two baselines. However, more related approaches could be included to strengthen the experiments (see below for details).


Weaknesses:
- The paper studies domain generalization and yet fails to position it in the right literature. By a simple search of "domain generalization" using Google Scholar, I found several existing works on this problem and have listed some below. The authors may consider to include them in both the related works and the experiments.

Questions:
1. It is intuitive to directly define the data augmentation by x_i+Grad_x J_d. Why is it necessary to instead define it as the inverse transformation G^{-1}(g') and then go through the approximations to derive the final augmentation?
2. Is the CrossGrad training necessary? What if one trains the network in two steps? Step 1: learn G using J_d and a regularization to avoid misclassification over the labels using the original data. Step 2: Learn the classification network (possibly different from G) by the domain-dependent augmentation.


Saeid Motiian, Marco Piccirilli, Donald A. Adjeroh, and Gianfranco Doretto. Unified deep supervised
domain adaptation and generalization. In IEEE International Conference on Computer
Vision (ICCV), 2017.

Muandet, K., Balduzzi, D. and Schölkopf, B., 2013. Domain generalization via invariant feature representation. In Proceedings of the 30th International Conference on Machine Learning (ICML-13) (pp. 10-18).

Xu, Z., Li, W., Niu, L. and Xu, D., 2014, September. Exploiting low-rank structure from latent domains for domain generalization. In European Conference on Computer Vision (pp. 628-643). Springer, Cham.

Ghifary, M., Bastiaan Kleijn, W., Zhang, M. and Balduzzi, D., 2015. Domain generalization for object recognition with multi-task autoencoders. In Proceedings of the IEEE international conference on computer vision (pp. 2551-2559).

Gan, C., Yang, T. and Gong, B., 2016. Learning attributes equals multi-source domain generalization. In Proceedings of the IEEE Conference on Computer Vision and Pattern Recognition (pp. 87-97).

---

> ### Public Comment · (anonymous) · 2017-12-10
> **Nice paper - More missing related work.**
>
> There is one recent DG paper that seems to have related methodology.
> Li et al, AAAI 2018, Learning to Generalize: Meta-Learning for Domain Generalization.
> ( https://arxiv.org/abs/1710.03463 )
> It would be good to contrast this as well.

---

> ### Author Response · Authors · 2017-12-27
> **Response to reviewer**
>
> We thank the reviewer for their time and effort.
>
> > It is intuitive to directly define the data augmentation by x_i+Grad_x J_d. Why is it necessary to instead define it as the inverse transformation G^{-1}(g') and then go through the approximations to derive the final augmentation?
>
> Yes, computationally they turn out to be the same but our exposition provides some insight on why perturbing x by Grad_x J_d should provide generalization along domain.  Also, we hope this more flexible framework will inspire future work on alternative ways of sampling g-s and the corresponding inverse. We will add a discussion of both ways of motivating the perturbation (g from x, and x from g) in our revised draft.
>
> > Is the CrossGrad training necessary? What if one trains the network in two steps? Step 1: learn G using J_d and a regularization to avoid misclassification over the labels using the original data. Step 2: Learn the classification network (possibly different from G) by the domain-dependent augmentation.
>
> We implemented the suggested method and found that it performs worse than the baseline. The accuracy for label classification on the Google Fonts dataset, on the test set, is around .63 while the baseline is around .68. If we learn G as a separate first step using training domains, there is no guarantee that G will generalize in a “meaningful way” across unseen domains. Then using this G for data augmentation will not be helpful. CrossGrad tries to force G to learn a meaningful continuous domain representation using perturbation in the label space.
>
> > The paper studies domain generalization and yet fails to position it in the right literature. By a simple search of "domain generalization" using Google Scholar, I found several existing works on this problem and have listed some below. The authors may consider to include them in both the related works and the experiments.
>
> We thank the reviewer for pointing us to these references. Among these, we’ve already cited Motiian et al.’s work on domain generalization and used their model for comparison on the MNIST task. The other references will be discussed in the related work section in our revised draft. Many previous approaches try to erase domain information.  For instance, Muandet et al. '13  and Ghifary et al. '15 try to extract generalizable features across domains. Our model is different in that we try to leverage the information that domain features provide about labels.

---

> ### Comment · AnonReviewer1 · 2018-01-17
> **Post-rebuttal comments**
>
> The rebuttal addresses my questions. The authors are recommended to explicitly use "domain generalization" in the paper and/or the title to make the language consistent with the literature.

---

### Official Review · AnonReviewer3 · 2017-11-30
**The authors introduce the CROSSGRAD method to address the problem of  multi-domain data to adaptation used to learn a classifier that generalizes to new domains. The  experimental validation on benchmark datasets shows that the proposed 'data augmentation' using multi-domain data leads to improved performance compared to traditional approaches for the task.**

**Rating:** 7
**Confidence:** 4

**Review:**

The method is posed in the Bayesian setting, the main idea being to achieve the data augmentation through domain-guided perturbations of input instances. Different from traditional adaptation methods, where the adaptation step is applied explicitly, in this paper the authors exploit labeled instances from several domains to collectively train a system that can handle new domains without the adaptation step. While this is another way of looking at domain adaptation, it may be misleading to say 'without' adaptation step. By the gradient perturbations on multi-domain training data, the learning of the adaptation step is effectively done. This should be clarified in the paper. The notion of using 'scarce' training domains to cover possible choices for the target domain is interesting and novel. The experimental validation should also include a deeper analysis of this factor: how the proposed adaptation performance is affected by the scarcity of the training multi-domain data. While this is partially shown in Table 8, it seems that by adding more domains the performance is compromised (compared to the baseline) (?).  It would be useful to see how the model ranks the multiple domains in terms of their relatedness to the target domain. Figs 6-7 are unclear and difficult to read. The captions should provide more information about the main point of these figures.

---

> ### Author Response · Authors · 2017-12-27
> **Response to reviewer**
>
> We thank the reviewer for their time and effort.
>
> > While this is another way of looking at domain adaptation, it may be misleading to say 'without' adaptation step. By the gradient perturbations on multi-domain training data, the learning of the adaptation step is effectively done. This should be clarified in the paper.
>
> The word “adaptation” suggests modifying the model for a given target domain. In contrast, we focus on domain-generalization, where we are not given any specific target domain. But the reviewer is right in that there is an implicit adaptation component, which we shall clarify in the revised draft.
>
> > The notion of using 'scarce' training domains to cover possible choices for the target domain is interesting and novel. The experimental validation should also include a deeper analysis of this factor: how the proposed adaptation performance is affected by the scarcity of the training multi-domain data. While this is partially shown in Table 8, it seems that by adding more domains the performance is compromised (compared to the baseline) (?).
>
> We shall add a deeper analysis/discussion in the revision. With training data covering a larger number of domains, the baseline automatically tends to become domain-aware, and domain generalization techniques (ours and the others) have less room for improvement.  However, it is possible that we can improve on our current performance by tuning the network parameters for different levels of scarcity. We shall explore this.
>
> > It would be useful to see how the model ranks the multiple domains in terms of their relatedness to the target domain.
>
> The analysis in Figure 6 is motivated by exactly this question. Instead of giving a single number (or rank), we have visualized the relation between the training and test domains in terms of (projections of) the “g” embedding. We could repeat these plots for other domains like Font, though these are probably best presented in supplementary material.
>
> > Figs 6-7 are unclear and difficult to read. The captions should provide more information about the main point of these figures
>
> We will address this in our revised draft.

---

### Official Review · AnonReviewer2 · 2017-12-02
**A novel domain-robust training method**

**Rating:** 8
**Confidence:** 4

**Review:**

Quality, clarity : Very well written, well motivated, convincing experiments and analysis
Originality: I think they framed the problem of domain-robustness very well: how to obtain a "domain level embedding" which generalizes to unseen domains. To do this the  authors introduce the CrossGrad method, which trains both a label classification task and a domain classification task (from which the domain-embedding is obtained)
Significance: Robustness in new domains is a very important practical and theoretical issue.

Pros:
- It's novel, interesting, well written, and appears to work very well in the experiments provided.

Cons:
- Formally, for the embedding to generalize one needs to make the "domain continuity assumption", which is not guaranteed to hold in any realistic settings (e.g. when there are no underlying continuous  factors)
- The training set needs to be in the form (x,y,d) where 'd' is a domain, this information might not exist or be only partially present.
- A single step required 2 forward and 2 backward passes - thus is twice as expensive.

Constructive comments:
- Algorithm 1 uses both X_l and X_d, yet the text only discusses X_d, there is some symmetry, but more discussion will help.
- LabelGrad is mentioned in section 4 but defined in section 4.1, it should be briefly defined in the first mention.

---

> ### Author Response · Authors · 2017-12-27
> **Response to reviewer**
>
> We thank the reviewer for their time and effort.
> > 'Formally, for the embedding to generalize one needs to make the "domain continuity assumption", which is not guaranteed to hold in any realistic settings (e.g. when there are no underlying continuous  factors)'
>
> Yes, real-life domains are mostly discrete (e.g. fonts, speakers, etc) but their variation can often be captured via latent continuous features (e.g. slant, ligature size, etc. for fonts; and speaking rate, pitch, intensity, etc. for speech).  CrossGrad strives to characterize these continuous features for capturing domain variation.
>
> > 'The training set needs to be in the form (x,y,d) where 'd' is a domain, this information might not exist or be only partially present.'
>
> We do not need domain information for all the data that will be used in our eventual training: we can bootstrap from a relatively small amount of domain-labeled data, by training a  classifier for the domains present in the training data, which can then be used to label the rest of the data. We also note that the domain adaptation/generalization literature typically does rely on training data with source-domain labels.
>
> We are fixing the revision with other helpful comments by the reviewer on notation and writing.

---

### Official Review · AnonReviewer4 · 2017-12-03
**A novel data augmentation method guided by domain specific information**

**Rating:** 7
**Confidence:** 5

**Review:**

The authors proposed to perturbed the estimated domain features for data augmentation, which is done by using the gradients of label and domain classification losses. The idea is interesting and new. And the paper is well written.

My major conerns are as follows:
1. Section 3 seems a bit too lengthy or redundant to derive the data augmentation by introducing the latent domain features g. In fact, without g, it also makes sense to perturb x as done in lines 6 and 7 in Alg. 1.
2. The assumption in (A1) can only be guaranteed under certain theoretical conditions. The authors should provide more explanations to better convey the assumption to readers.

Minors:
1. LabelGrad was not defined when firstly being used in Section 4.
2. Fig. 1 looks redundant.

---

> ### Author Response · Authors · 2017-12-27
> **Response to reviewer**
>
> We thank the reviewer for their time and effort.
>
> > Section 3 seems a bit too lengthy or redundant to derive the data augmentation by introducing the latent domain features g. In fact, without g, it also makes sense to perturb x as done in lines 6 and 7 in Alg. 1. 2.
>
> A similar concern was raised by Reviewer 1 as well. We wanted to provide some insight on why perturbing x by Grad_x J_d should provide generalization along domain.  Also, we hope this more flexible framework will inspire future work on alternative ways of sampling g-s and the corresponding inverse.
>
> > The assumption in (A1) can only be guaranteed under certain theoretical conditions. The authors should provide more explanations to better convey the assumption to readers.
>
> Yes, but in many cases, domain variation can be captured via latent continuous features (e.g. slant, ligature size, etc. for fonts; and speaking rate, pitch, intensity, etc. for speech).  CrossGrad strives to characterize these continuous features for capturing domain variation. We shall elaborate on this further in our revised draft.
>
> > Minors: 1. LabelGrad was not defined when firstly being used in Section 4. 2. Fig. 1 looks redundant.
>
> We will fix the issues pointed out .

---

### Author Response · Authors · 2018-01-05
**Revision**

We have uploaded a revised draft of our paper, in line with the valuable comments. To summarize the changes:
- A discussion of the additional papers referenced in the reviews has been added to the Related Work section.
- A comparison with Ghifary et al. 2015 has been added to Table 9.
- The domain continuity assumption is elaborated on p3-4.
- The domain-based perturbation for the label classifier is motivated in the "forward" direction, i.e. directly as \nabla_x J_d (and similarly for the domain classifier). Reviewers felt this was more intuitive. To give insight into the relation between the perturbations of input x and domain features g, we added a short paragraph at the end of p4 and moved the related "backward" derivation (the one in the original draft) to an appendix.
- A note on how the base networks are modified and how the complementary loss is used has been added.
- Several other small issues in the exposition have been fixed.

We thank the reviewers and other commenters for their suggestions in regards to improving the exposition, and look forward to further suggestions and comments.

---

### Decision · Program_Chairs · 2018-01-29
**ICLR 2018 Conference Acceptance Decision**

**Decision:**

Accept (Poster)

**Comment:**

Well motivated and well received by all of the expert reviewers. The AC recommends that the paper be accepted.